# Controlling Tunneling Characteristics via Bias Voltage in Bilayer Graphene/WS_2_/Metal Heterojunctions

**DOI:** 10.3390/nano12091419

**Published:** 2022-04-21

**Authors:** Zongqi Bai, Sen Zhang, Yang Xiao, Miaomiao Li, Fang Luo, Jie Li, Shiqiao Qin, Gang Peng

**Affiliations:** 1College of Liberal Arts and Sciences, National University of Defense Technology, Changsha 410073, China; baizongqi17@nudt.edu.cn (Z.B.); lijie479@nudt.edu.cn (J.L.); sqqin8@nudt.edu.cn (S.Q.); 2College of Advanced Interdisciplinary Studies & Hunan Provincial Key Laboratory of Novel Nano-Optoelectronic Information Materials and Devices, National University of Defense Technology, Changsha 410073, China; xiaoyang624@nudt.edu.cn (Y.X.); limiaomiao20a@nudt.edu.cn (M.L.); luofang@nudt.edu.cn (F.L.)

**Keywords:** field-effect tunneling transistors, graphene-based heterojunctions, FN tunneling, energy band diagrams

## Abstract

Van der Waals heterojunctions, formed by stacking two-dimensional materials with various structural and electronic properties, opens a new way to design new functional devices for future applications and provides an ideal research platform for exploring novel physical phenomena. In this work, bilayer graphene/WS_2_/metal heterojunctions (GWMHs) with vertical architecture were designed and fabricated. The tunneling current–bias voltage (*I*_t_ − *V_b_*) properties of GWMHs can be tuned by 5 × 10^6^ times in magnitude for current increasing from 0.2 nA to 1 mA with applied bias voltage increasing from 10 mV to 2 V. Moreover, the transfer properties of GWMHs exhibit n-type conduction at *V_b_* = 0.1 V and bipolar conduction at *V_b_* = 2 V; these findings are explained well by direct tunneling (*DT*) and Fowler–Nordheim tunneling (*FNT*), respectively. The results show the great potential of GWMHs for high-power field-effect transistors (FETs) and next-generation logic electronic devices.

## 1. Introduction

Graphene FETs have the advantages of high carrier mobility and large current density [1,2,3,4,5,6], but their low on/off ratio is a significant disadvantage. On the other hand, FETs based on transition metal dichalcogenides (TMDCs) exhibit a high on/off ratio [7,8,9,10,11,12], but the current density in the on-state is greatly limited by the Schottky barrier between the metal and TMDCs [13,14]. Hence, FETs based on pure graphene or pure TMDCs cannot meet the demands of device applications or the curiosity of researchers. Van der Waals heterostructures, fabricated by accurately stacking two-dimensional (2D) materials, exhibit a series of distinctive physical phenomena and properties [15,16,17,18,19,20,21]. As a result, the Van der Waals heterostructure is considered a promising candidate for electronic and photoelectric devices in the post-silicon era [15,21]. In the family of Van der Waals heterostructures, the Van der Waals tunneling heterojunctions consisting of graphene-2D material metal have attracted the most attention because of their fascinating physical properties and wide-ranging applications in tunneling spread field-effect transistors [18,19,20,21], photodetectors [22,23,24,25,26], and magnetic tunnel heterojunctions [27,28].

Among the device applications of Van der Waals heterojunctions, tunneling transistors based on the carrier tunneling effect have particular significance due to their advantage of a large on/off ratio and the associated promising prospect of low power consumption. To achieve a sufficiently high on-state current density and a very low off-state current density in graphene-based tunneling transistors, a suitable tunneling layer should be delicately designed. Among 2D materials, hexagonal boron nitride (h-BN) is widely used as the tunneling layer [29,30] because of its larger bandgap (5.6–6 eV), through which the thermionic current in h-BN-based tunneling devices can be tightly controlled [18,19,20]. However, the electron affinity of h-BN (2~2.3 eV) is much lower than the work function of graphene (~4.6 eV) [31] and common electrode metals, such as Au (~5.1 eV). Therefore, the barrier height for electron tunneling through the h-BN/graphene (or electrode metal Au) is 2.3–2.6 eV (or 2.8~3.1 eV), which is high and limits the on-state current density [13,14]. Furthermore, the Fermi level of graphene can be tuned up or down (±0.3 eV) from the Dirac point (which is the point where the conduction and valence bands of graphene join) by applying gate voltages [21], which are significantly lower than the bandgap voltage of h-BN. Therefore, it is difficult to adjust the Fermi level of the graphene away from the bandgap of h-BN, and graphene’s advantage of gate-voltage controllability is thus frustrated in h-BN/graphene-based devices. As a result, new 2D materials with a suitable bandgap are highly desired as the interlayer.

Tungsten disulfide (WS_2_) is an n-type semiconductor [11], and it has a bandgap ranging from 2 eV for a single layer to 1.3 eV in bulk [32,33]. The electron affinity of WS_2_ is 4.2 eV [32], which is quite close to graphene’s work function (~4.6 eV). As a result, we can tune the Fermi level of the graphene into or out of the conduction band of WS_2_, taking advantage of graphene’s gate-voltage controllability. Hence, WS_2_ is an ideal material for investigating the tunneling mechanism and improving the performance of graphene-based tunneling devices [21].

In this study, bilayer graphene–WS_2_–metal vertical heterojunctions (GWMHs) were fabricated via a controlled dry transfer method (see details in Methods) [34]. Bilayer graphene was employed as the bottom layer to avoid the influence of SiO_2_ substrate on the monolayer graphene. Moreover, Cr/Au was selected as the top metal because Cr’s work function (~4.6 eV) [35] is close to that of graphene at zero bias. In this regard, we can achieve Ohmic contact at the graphene–Cr/Au interface and Schottky contact between the WS_2_ barrier and the Cr/Au electrode with a relatively low barrier, resulting in a high on-state current density (4 × 10^7^ A m^−2^) and a high on/off ratio (5 × 10^6^) by sweeping the bias voltages. More interestingly, the GWMHs device exhibits n-type conduction at 0.1 V bias voltage and bipolar conduction at 2 V bias voltage, which is unusual in semiconducting devices but can be explained well by direct tunneling (*DT*) and Fowler–Nordheim tunneling (*FNT*), respectively. The results of this study not only pave the way to polarity-controllable high-performance 2D transistors but also provide an ideal paradigm for designing next-generation logic electronic devices.

## 2. Materials and Methods

**Device fabrication.** Few-layer bottom graphene and a WS_2_ tunneling barrier layer were mechanically exfoliated from bulk graphite and WS_2_ single crystals from the HQ Graphene company (Groningen, The Netherlands) [30]. After exfoliation, the bottom graphene layer was transferred onto the SiO_2_/Si substrate, and the exfoliated WS_2_ layer was transferred onto a polydimethylsiloxane (PDMS) membrane. Then, the vertical graphene–WS_2_ Van der Waals heterostructures were carefully assembled with the dry transfer method [34] using a custom-built transfer stage. Finally, the graphene and WS_2_ layers were deposited with Cr/Au electrodes using standard e-beam lithography and an e-beam evaporation process. The deposited film thicknesses of Cr and Au electrodes were 5 nm and 50 nm, respectively.

**Structure characterization.** The Raman spectra of graphene and WS_2_ layers were measured by a confocal Raman spectrometer (Witec Alpha 300R, Ulm, Germany) at room temperature in ambient conditions. The excitation wavelength was 532 nm. The laser spot was about 1 μm in diameter and had a power of 1 mW. The thickness of the WS_2_ layer was determined by atomic force microscopy using the tapping mode (AFM, NT-MDT NTEGRA Prima, Zelenograd, Russia).

**Electrical transport measurements.** The tunneling *I*–*V* characteristics of our devices were measured by a low-temperature probe station equipped with source meters and lock-in amplifiers. The samples were placed in a vacuum chamber (<10^−6^ Torr) with a variable temperature from 300 to 5 K.

## 3. Results and Discussion

Both the graphene and WS_2_ layers were mechanically exfoliated from high-quality bulk crystals and then vertically stacked layer-by-layer via a dry transfer method to form the heterostructures. The bottom graphene layer was electrically attached to Cr/Au metal electrodes, and the whole heterostructure device was placed on an Si substrate with a 290 nm SiO_2_ layer on top (for details of device fabrication, please see the Methods section). Figure 1a depicts the schematics of the GWMHs tunneling device. Figure 1b shows the optical image of the tunneling device, in which the overlapped GWMHs sandwich area is about 25 μm^2^. The thicknesses of the graphene bottom layer and the WS_2_ middle layer were measured by atomic force microscopy (AFM) and were about 0.89 and 6.4 nm (corresponding to 2 layers for graphene and 9–10 layers for MS_2_, respectively) at one edge of the device, as shown in the Figure 1b inset. Figure 1c shows the scanning Raman map of the integrated intensity of Si (peak position: 520 cm^−1^, integration width: 20 cm^−1^). The shapes of the graphene and WS_2_ layers are clearly visible, correlating well with the optical image. Raman spectroscopy was used to characterize the number of layers of WS_2_ and graphene, as shown in Figure 1d,e. The Raman spectrum of WS_2_ exhibits two characteristic peaks—the *E*_1g_^2^ peak at 350 cm^−1^ and the *A*_1g_ peak at 420 cm^−1^—which are consistent with the previous Raman studies of few-layer WS_2_ [36]. The number of layers of graphene can also be determined as a bilayer by the Raman spectra according to the shape of the 2D modes [2,37,38,39].

After structural characterization, electrical measurements of the tunneling current density–bias voltage (*j*_t_ − *V_b_*) characteristics were performed in the GWMHs devices. By applying a bias voltage (*V_b_*) between the bottom graphene layer and the top Cr/Au electrode, the tunneling current density (*j*_t_) through the multilayer WS_2_ could be detected. Figure 2a shows the *j*_t_ − *V_b_* curves under various gate voltages (*V*_g_) at 300 K, and Figure 2b shows the map of *j*_t_ as a function of the bias voltage *V_b_* and the gate voltage *V*_g_ at 300 K. Figure 2c,d show the characteristic *j*_t_ − *V_b_* curves and the *j*_t_(*V_b_*,*V*_g_) map at 5 K, respectively. As illustrated in Figure 2c, the tunneling current density (*j*_t_) of GWMHs can be tuned by a factor of 5 × 10^6^, e.g., from 8 A m^−2^ to 4 × 10^7^ A m^−2^, with the bias voltage changing from 10 mV to 2 V at 5 K. Under a very small bias at 10 mV, current leakage still exists with magnitude 8 A m^−2^, resulting from host point defects in the WS_2_ layer [40]. Figure 2e illustrates the current density of the GWMHs in a logarithmic scale as a function of bias voltages under a gate voltage of −60 V at different temperatures. If the off-state current (I-OFF) and on-state current (I-ON) are defined as currents with biases of 10 mV and 2 V, respectively, then the current on/off ratio can be calculated, as shown in Figure 2f. As the temperature increased from 5 to 300 K, the current on/off ratio decreased from 5 × 10^6^ to 6 × 10^4^ because of the increased current leakage at higher temperatures [40]. Compared with the devices based on graphene/h-BN/metal heterojunctions that achieve a lower on-state current density (10^6^ A m^−2^ at 25 V bias) [19], our GWMHs devices achieved a higher on-state current density (4 × 10^7^ A m^−2^ at 2 V bias).

As depicted in Figure 3a,d, the transfer curves (*j*_t_−*V*_g_) exhibit typical n-type conduction at a low bias (*V_b_* = 0.1 V) but bipolar conduction at a high bias (*V_b_* = 2 V). By changing the gate voltage, the Fermi level of bottom graphene can be tuned. Figure 3b,c shows the band diagrams of the GWMHs heterostructure at 0.1 V bias. For a gate voltage at −60 V, the Fermi levels of the bottom graphene layers align with the bandgap of WS_2_, but the Fermi level of the top metal electrodes cannot be tuned because of their high DOS, as shown in Figure 3b. The tunneling barrier was obviously trapezoid-shaped by the bias-induced electric field penetrating through the bottom graphene layer and the top metal electrodes. In this area, the total resistance of the GWMHs device was dominated by the direct tunneling process, while the conduction was limited by the relatively large barrier height and width. As the gate voltage increased, the Fermi level of graphene moved above the conduction band of WS_2,_ lowering the tunneling barrier and increasing the tunneling current, as shown in Figure 3c. Therefore, the GWMHs devices showed n-type transfer characteristics at a 0.1 V bias. When the bias voltage was set to 2 V, the tunneling barrier was reshaped to a triangle, and the effective width of the barrier was significantly reduced, which resulted in a larger tunneling current density, as shown in Figure 3e,f. Under this condition, the total resistance was no longer dominated by the tunneling resistances, but instead by the resistances of the graphene electron. As shown in Figure 3d, the transfer curve exhibited clear bipolar characteristics. Compared with the insulator h-BN with a wide bandgap, the graphene tunneling devices exhibited a high tunneling current and controllable conduction polarity, which is uncommon in semiconducting devices.

In the graphene–WS_2_–metal heterojunction device, it is possible to tune the Fermi level of the bottom graphene and tailor the shape of the tunneling barrier, providing a platform for exploring the tunneling mechanism under different bias voltages. At low bias voltages, direct tunneling occurs, and the tunneling barrier width is equal to the WS_2_ barrier thickness, which is unaffected by the bias and gate voltages. The direct tunneling current density *j_DT_* can be approximated by the following equation [41,42]:(1)$$j_{DT}=\frac{\sqrt{m\varphi_{B}}e^{2}V_{b}}{h^{2}d}\;\exp\left[\frac{-4\pi \sqrt{m^{*}\varphi_{B}}d}{h}\right]$$

The WS_2_ barrier can be further shaped by higher bias voltages, converting it from a trapezoid to a triangle. In this circumstance, the tunneling barrier width is no longer equal to the WS_2_ barrier thickness, which is reduced as *V_b_* is increased. Therefore, the tunneling probability is markedly increased, and the resistance of bottom graphene dominates the total resistance. Fowler–Nordheim tunneling (FN tunneling) occurs, which can be described by the following equation [41,42]:(2)$$j_{FNT}=\frac{e^{3}mV_{b}^{2}}{8\pi h\varphi_{B}d^{2}m^{*}}\;\exp\left[\frac{-8\pi \sqrt{2m^{*}}\varphi_{B}^{3/2}d}{3heV_{b}}\right]$$

In Equations (1) and (2), *m* and *m** are the masses of free and effective electrons, respectively. $\varphi_{B}$ and *d* are the effective barrier height and width, respectively, determined by the difference between graphene’s Fermi level and the bottom of WS_2_’s conduction band at the graphene–WS_2_ interface, respectively. *h* is Planck’s constant, and *e* is the unit charge of the electron.

Because of the high electric field generated during the FN tunneling process, the Joule heating effect frequently causes a large thermionic emission current over the triangle-shaped barrier. To suppress the thermionic current and determine the effective barrier height $\varphi_{B}$, we measured *j*_t_ − *V_b_* characteristics under different back-gate voltages at a temperature of 5 K. The Equation (2) can be transformed into [41,42]:(3)$$\ln\frac{j_{FNT}}{V_{b}^{2}}=\ln\frac{q^{3}m}{8\pi h\varphi_{B}d^{2}m^{*}}-\frac{8\pi \sqrt{2m^{*}}\varphi_{B}^{3/2}d}{3hqV_{b}}$$

Equation (3) helps investigate the *j*_t_ − *V_b_* curves and establish a reference point for defining the tunneling barrier height. Figure 4a–e shows ln(|*j_t_*|/*V_b_*^2^) as a function of 1/*V_b_* in the GWMHs tunneling device at different *V*_g_ values at 5 K. At a higher *V_b_*, the larger negative slope indicates that the FN tunneling dominated the carrier transport and current flow in this region, while at lower *V_b_*, direct tunneling of charge carriers through the ultrathin WS_2_ layer dominated. If the effective mass value of the electron (*m**) is set at 0.3 m [43], then the barrier height $\varphi_{B}$ for the FN tunneling can be extracted from Figure 4a–e. $\varphi_{B}$ can be determined from the negative slope k by Equation (4) [41,42]:(4)$$k=-\frac{8\pi \sqrt{2m^{*}}\varphi_{B}^{3/2}d}{3hq}$$

Figure 4f depicts the barrier height for FN tunneling. $\varphi_{B}$ increases as the *V*_g_ changes from zero to negative values (hole doping). The gate-tunability of the carrier tunneling regions and the barrier height in our GWMHs devices opens a new road to control the characteristics of vertical field-effect tunneling transistors. In addition, with the continuous progress of lithography technology, lateral tunneling heterojunctions have also attracted much attention [44,45,46]. Heterostructures of two-dimensional materials offer a plethora of opportunities in materials science, condensed-matter physics, and device engineering.

## 4. Conclusions

In summary, a tunable field-effect tunneling transistor based on graphene–WS_2_–metal heterojunctions (GWMHs) was studied in this work. The tunneling current density (*j*_t_) of GWMHs changes by a factor of 5 × 10^6^, with the applied bias voltages changing from 10 mV to 2 V, and a high on-state current density (4 × 10^7^ A m^−2^) was achieved. Moreover, the transfer properties of GWMHs exhibit n-type conduction at a 0.1 V bias and bipolar conduction at 2 V, which are well-explained by direct tunneling (*DT*) and Fowler–Nordheim tunneling (*FNT*), respectively. Our results show great potential for GWMHs in high-power FET devices and next-generation logic electronic devices.

## Figures and Tables

**Figure 1 nanomaterials-12-01419-f001:**
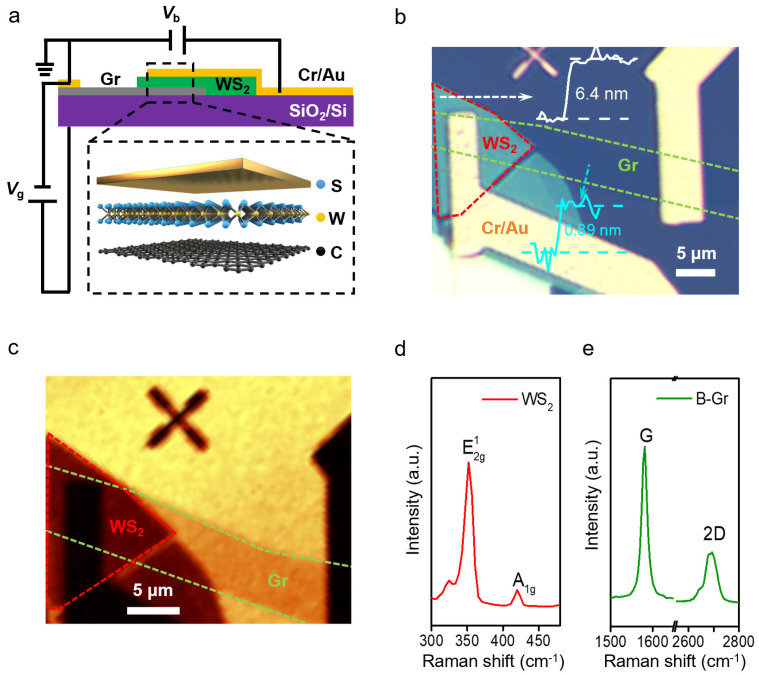
Characterizations of the GWMHs tunneling devices. (**a**) Schematics of the GWMHs tunneling device with the crystalline structure of each layer shown in an enlarged view. The WS_2_ layer is sandwiched by the graphene and metal electrode. Bias voltage (*V_b_*) is applied between the bottom graphene and the top Cr/Au electrode; gate voltage (*V*_g_) is applied between the bottom graphene and the SiO_2_/Si substrate. (**b**) Optical microscopy image of the GWMHs tunneling device. The graphene and WS_2_ layers are marked with green and red dash lines. The inset white line and cyan line show the thicknesses of the WS_2_ layer and the graphene layer which were determined by AFM to be about 6.4 and 0.89 nm, respectively. (**c**) Scanning Raman mapping of integrated intensity of Si (peak position: 520 cm^−1^, integration width: 20 cm^−1^) showing the shapes of the heterojunction. The green and red dashed lines mark the position of the WS_2_ and graphene layers, respectively. (**d**,**e**) Raman spectra of WS_2_ and graphene, respectively.

**Figure 2 nanomaterials-12-01419-f002:**
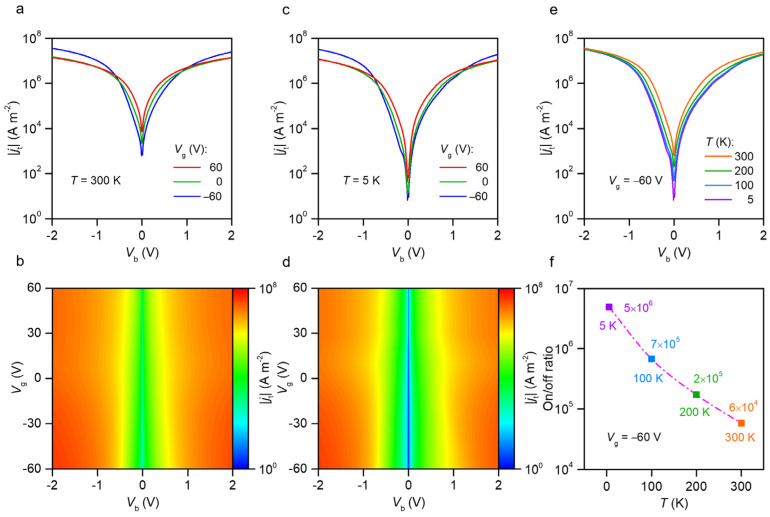
Tunneling current characteristics of GWMH devices. (**a**) Characteristic curves of tunneling current density versus bias voltage (*j*_t_ − *V_b_*) under different gate voltages *V*_g_ at 300 K. (**b**) Map of tunneling current density *j*_t_ as a function of bias voltage *V_b_* and gate voltage *V*_g_ at 300 K. (**c**,**d**) Characteristic *j*_t_ − *V_b_* curves and *j*_t_(*V_b_*,*V*_g_) map at 5 K, respectively. (**e**) Tunneling current density *j*_t_ versus bias voltage *V_b_* under gate voltage *V*_g_ =−60 V from 300 to 5 K. (**f**) Current on/off ratio under −60 V gate voltage from 5 to 300 K.

**Figure 3 nanomaterials-12-01419-f003:**
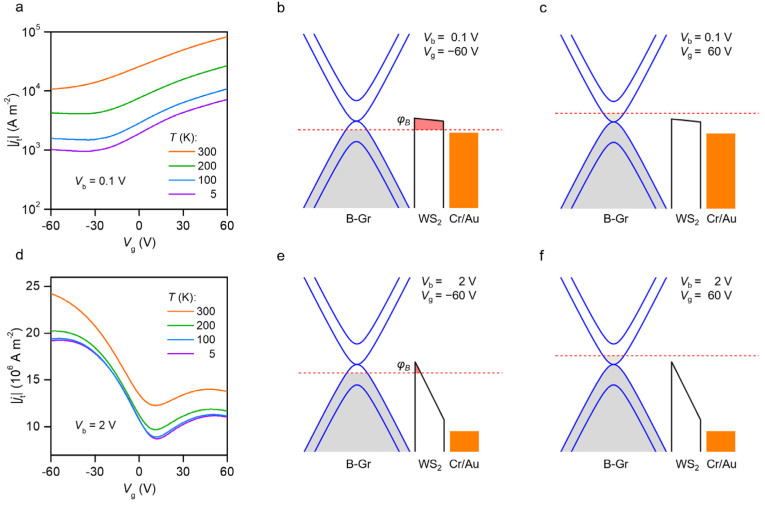
Controlling the carrier transport polarity in GWMH tunneling devices. (**a**–**c**) The transfer curves of the GWMHs device at low bias (*V_b_* = 0.1 V) and the corresponding energy band diagrams under *V*_g_ = −60 V and *V*_g_ = 60 V, respectively. (**d**–**f**) The transfer curves of the GWMHs device at low bias (*V_b_* = 2 V) and the corresponding energy band diagrams under *V*_g_ = −60 V and *V*_g_ = 60 V, respectively. The red dash line is the Fermi level of the bottom graphene.

**Figure 4 nanomaterials-12-01419-f004:**
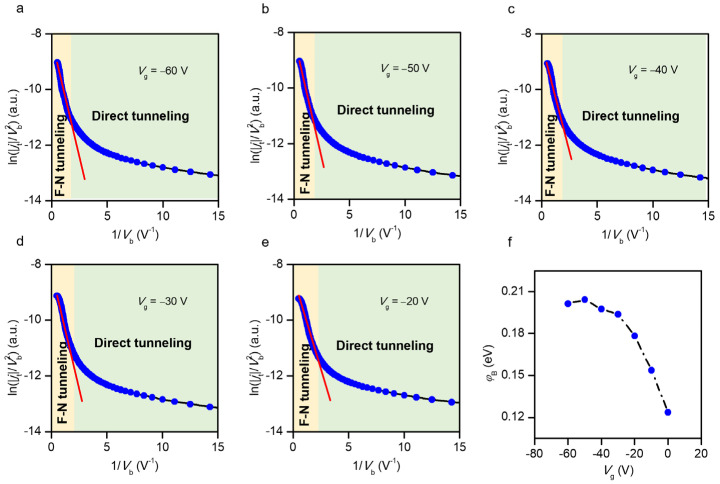
FN tunneling and direct tunneling in GWMHs devices. (**a**–**e**) ln(|*j*_t_|/*V_b_*^2^) versus 1/*V_b_* curves of the GWMHs devices at different values of *V*_g_. The linear behavior with FN tunneling is observed at a higher bias voltage (yellow region with red dashed line), and direct tunneling is observed at a lower bias voltage (green region). (**f**) The variation in the barrier height as a function of *V*_g_.

## Data Availability

Data sharing is not applicable to this article.

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
