# Peer review of "Controlling Tunneling Characteristics via Bias Voltage in Bilayer Graphene/WS2/Metal Heterojunctions"

_nanomaterials, 2022, doi:10.3390/nano12091419_

Round 1
Reviewer 1 Report
See attached document.

Reviewer 2 Report
The authors have reported tunneling characteristics that can be switched between direct tunneling or Fowler-Nordheim tunneling depending on the bias voltage from their bilayer graphene/WS2/metal heterojuctions. Moreover, in doing so, the conduction mechanism of the device can be tailored to be n-type or bipolar. The presented results can contribute towards the development of new and interesting logic electronic devices. Accordingly, I am glad to recommend the paper for publication in Nanomaterials. I am also suggesting adding a Figure illustrating the band alignment of graphene, WS2 and Cr, to help improve the introduction portion of the paper.
Reviewer 3 Report
In this manuscript, authors demonstrate the characteristics of a tunneling transport in a vertical stack of metal/WS2/Gr structure. Graphene is used as the bottom electrode, metal as the top electrode, and WS2 as the tunnel barrier. By the application of a voltage across the stack (labeled as Vb), the transport regime is changed from n-type (at low voltage) to bipolar (at high voltage). Large ON/OFF ratios are reported, 5e6 at 5K and 6e4 at 300K. In addition, tunability of the fermi level in graphene via gating is used as a control knob.
Although the topic might be a bit old fashion, the manuscript is well prepared, straightforward, and is easy to follow. Experiments are explained properly and contain interesting information. The demonstrate ON/OFF ratio is also decent! However, there are some important aspects of the study that needs further elaboration before publication of the manuscript can be granted. I can suggest the publication of manuscript only if the following points are addressed:
- Authors have not properly explained why a 6.4nm WS2 film is used as the tunnel barrier? Why not a thinner WS2? As authors have mentioned a monolayer WS2 has a larger band gap and can serve as a better tunnel barrier. this point needs to be elaborated better. Given the difficulties of the process, it is probably too much to expect having a range of thicknesses tested for comparison. But having at least another device with a thin WS2 tunnel barrier, maybe a few layers (1-3 layers), will be a nice addition.
- 2; What is the source of the current in the OFF state? there is leakage through the stack at zero bias (especially at room temperature (fig. 2a)). do authors know what is the leakage mechanism? This should be answered in two parts (a) Vb is set to 0, then what is the drive behind the current? Current requires a driving force. (b) what is the transport channel? Is it the structural defects and atomic pinholes within the WS2 that serve as channels for the current leakage, leading to the OFF-state current? transition metal dichalcogenides (WS2, MoS2, MoSe2) are known to host point defects (see for example ex; ACS Nano2018, 12, 12, 12795–12804). Please elaborate this point carefully.
- Combining above two points (thickness of the WS2 and possible defects and leakage); One might think that there is a limit in how thin the WS2 layer can be. Thin WS2 has a larger bandgap, but perhaps leaky in the off state because of the defect states in monolayer films, according to the reference I shared above.
- Figure 2e; reduction in the OFF current at low temperature is understandable. But what why the ON current is not decreasing at lower temperatures? Density of carriers as well as their mobility is expected to be temperature dependent. please elaborate.
- Figure 3(d-f); the bipolar process is perhaps stemming from the FN transport. This part is well explained in the manuscript; but why the current in negative Vg (corresponding to figure 3e) is larger than the current in the positive Vg (corresponding to the situation shown in panel f)? should it not be the other way around?
- A suggestion: when it comes to tunneling devices, most people naturally think of vertical heterostructures. But what about lateral (i.e., side-by-side) junctions? Is there any fundamental problem with using such structures for tunneling devices? Lateral junctions can be lithographically produced with high qualities and with controllable size and dimensions in predefined locations. Atomic quality interfaces have been demonstrated between various 2D materials (see a review of available techniques at: Optical Materials Express Vol. 9, Issue 4, pp. 1590-1607 (2019)). I highly recommend adding a brief session at the end and discuss the possibility of using lateral junctions for tunneling applications. I believe there is a huge potential that is simply being left unexplored.
Round 2
Reviewer 1 Report
The authors have satisfactorily answered my previous comments/claims, therefore I recommend the manuscript for publication. However, there are some typos and misspelt words that must be corrected first:
-line 17: increasement
-line 39: tunnelingspread
-line 54: fromthe
-line 60: WS2
Author Response
Response:
Thanks very much for pointing out these typos and misspelt words, we have revised the manuscript accordingly.
-line 17: increasement----”increasing”
-line 39: tunnelingspread ----” tunneling spread”
-line 54: fromthe ----” from the”
-line 60: WS2----”WS2”
Reviewer 3 Report
Authors have properly answered most of my comments with enough explanations and added new material to the manuscript. However, there is still one minor point that I recommend adding to the manuscript.
following my comment #2; I strongly suggest to add a sentence to the revised manuscript and clearly mention the voltage at which I(ON) and I(OFF) are measured for the data points shown in Figure 2e.
Paper does not need to go to be submitted back for my evaluation and upon the addition of this sentence, it shall be ready for publications.
Author Response
Response:
Thanks for the advice, and we absolutely agree on that point. We have added a sentence in the revised manuscript to point the voltage at which I(ON) and I(OFF) are measured, the sentence is as follow:
Line 131-136: “Figure 2(e) illustrates the current density of the GWMHs in a logarithmic scale as a function of bias voltages under gate voltage -60V at different temperature. If off-state current (I-OFF) and on-state current (I-ON) are defined as the current with a bias of 10 mV and 2 V, respectively, then the current on/off ratio could be calculated and shows in figure 2(f). With increasing temperature from 5K to 300K, the current on/off ratio de-creases from 5×106 to 6×104 because of the increased current leakage at higher temperature[40].”